# The Role of Intracellular Ca^2+^ and Mitochondrial ROS in Small Aβ_1-42_ Oligomer-Induced Microglial Death

**DOI:** 10.3390/ijms241512315

**Published:** 2023-08-01

**Authors:** Aiste Jekabsone, Silvija Jankeviciute, Katryna Pampuscenko, Vilmante Borutaite, Ramune Morkuniene

**Affiliations:** 1Neuroscience Institute, Lithuanian University of Health Sciences, LT-50162 Kaunas, Lithuania; aiste.jekabsone@lsmu.lt (A.J.); silvija.jankeviciute@lsmu.lt (S.J.); katryna.pampuscenko@lsmu.lt (K.P.); vilmante.borutaite@lsmu.lt (V.B.); 2Faculty of Pharmacy, Lithuanian University of Health Sciences, LT-50162 Kaunas, Lithuania

**Keywords:** Alzheimer’s disease, amyloid-β, microglia, NMDA receptors, mitochondrial ROS, cell death, glutamate

## Abstract

Alzheimer’s disease (AD) is the most common form of dementia worldwide, and it contributes up to 70% of cases. AD pathology involves abnormal amyloid beta (Aβ) accumulation, and the link between the Aβ_1-42_ structure and toxicity is of major interest. NMDA receptors (NMDAR) are thought to be essential in Aβ-affected neurons, but the role of this receptor in glial impairment is still unclear. In addition, there is insufficient knowledge about the role of Aβ species regarding mitochondrial redox states in neurons and glial cells, which may be critical in developing Aβ-caused neurotoxicity. In this study, we investigated whether different Aβ_1-42_ species—small oligomers, large oligomers, insoluble fibrils, and monomers—were capable of producing neurotoxic effects via microglial NMDAR activation and changes in mitochondrial redox states in primary rat brain cell cultures. Small Aβ_1-42_ oligomers induced a concentration- and time-dependent increase in intracellular Ca^2+^ and necrotic microglial death. These changes were partially prevented by the NMDAR inhibitors MK801, memantine, and D-2-amino-5-phosphopentanoic acid (DAP5). Neither microglial intracellular Ca^2+^ nor viability was significantly affected by larger Aβ_1-42_ species or monomers. In addition, the small Aβ_1-42_ oligomers caused mitochondrial reactive oxygen species (mtROS)-mediated mitochondrial depolarization, glutamate release, and neuronal cell death. In microglia, the Aβ_1-42_-induced mtROS overproduction was mediated by intracellular calcium ions and Aβ-binding alcohol dehydrogenase (ABAD). The data suggest that the pharmacological targeting of microglial NMDAR and mtROS may be a promising strategy for AD therapy.

## 1. Introduction

Alzheimer’s disease (AD) is the most common cause of dementia in older adults, with an estimated 55 million cases worldwide, which are predicted to reach 150 million by 2050 [1]. AD pathogenesis involves the abnormal accumulation of amyloid beta (Aβ) peptides, and the action mechanism of various Aβ_1-42_ assemblies is of major interest. The neurotoxicity of Aβ_1-42_ oligomers includes the neuronal N-methyl D-aspartate receptor (NMDAR)-dependent disruption of Ca^2+^ homeostasis, the destabilization of the neuronal cell membrane [2,3], the promotion of inflammatory reactions [4,5], or mitochondrial injury [6,7]. Most studies refer to neuronal NMDAR as the primary target in the Aβ damage pathway, yet the role of NMDAR in the microglia-related events in AD remains unclear. We have previously reported that small Aβ_1-42_ oligomers (but not larger aggregates or monomers) induce neuronal and microglial plasma membrane depolarization and neuronal death; however, only microglial depolarization was prevented by the NMDAR blocker MK801 [8]. Recently, it has been shown that NMDA stimulates microglial proliferation and morphological transformation toward an activated state that is characterized by the increased expression of inducible nitric oxide synthase (iNOS), NO production, proinflammatory cytokine secretion, and reactive oxygen species (ROS) production [9,10,11]. Furthermore, the redox state directly controls NMDAR, and oxidative stress causes microglial activation, perineural net degradation, and impairs NMDARs [12,13]. On the other hand, NMDAR activation may induce superoxide generation through nicotinamide adenine dinucleotide phosphate (NADPH) oxidases, thereby leading to neuronal damage during anoxia/reoxygenation [14]. The overproduction of neuronal mitochondrial ROS (mtROS) triggers the activation of NMDARs and AMPARs, thereby increasing cytosolic calcium and neuronal death in frontotemporal dementia pathology [15]. In neurons, mtROS might be induced by Aβ-binding alcohol dehydrogenase (ABAD) [16,17]; therefore, targeting the Aβ-ABAD interaction has emerged as a novel therapeutic strategy for AD [18]. In addition, the mtROS-related inhibition of the respiratory chain complex I/II by rotenone and antimycin A markedly reduced the microglial capacity related to phagocytose Aβ [19]. The data suggest that microglial NMDARs, mitochondrial dysfunction, and mtROSs could likely be involved in Aβ-mediated neurotoxicity; however, the impact of differently oligomerized Aβ_1-42_ species on these targets is not consistently studied. Therefore, in this study, we investigated whether Aβ_1-42_ in the form of small (z-high < 5 nm) and large (z-high > 5 nm) oligomers, insoluble fibrils, and monomers exerted neurotoxic effects via changes in microglial intracellular Ca^2+^, NMDAR activation, and mtROSs, and if so, whether these events were interrelated.

## 2. Results

### 2.1. The Role of NMDARs in Aβ_1-42_-Induced Microglial Cytoplasmic Calcium Increase and Death

First, we investigated whether Aβ_1-42_ species of different aggregation states might cause changes in the cytosolic calcium ion concentration in primary microglial cultures (Figure 1). After 15 min of incubation of the cultured microglial cells with small Aβ_1-42_ oligomers, the level of intracellular Ca^2+^-dependent fluorescence increased by 18% compared with the control, and after 45 and 90 min, the fluorescence levels rose to 36% and 50%, respectively (Figure 1b,c,i). In contrast, the level of intracellular Ca^2+^ in microglia remained unchanged during a 90-minute treatment with Aβ_1-42_ monomers, large oligomers, and fibrils.

Next, we evaluated whether small Aβ_1-42_ oligomers induced microglial death. As shown in Figure 2a, in pure microglial cultures, small Aβ_1-42_ oligomers caused rapid necrosis in a concentration- and time-dependent manner: after 30 min of incubation with 0.5 μM and 1 μM small Aβ_1-42_ oligomers, the levels of necrotic (PI-positive) cells were about 10% and increased gradually over 4.5 h of incubation to eventually reach 22% and 61% necrosis at 0.5 μM and 1 μM concentrations, respectively. After 24 h, about 90% of the cells were necrotic in the 1 μM small Aβ_1-42_ oligomer-treated primary microglial cultures (Figure 2a,b); in the 0.5 μM small Aβ_1-42_ oligomer-treated primary microglial cultures, about 60% necrosis was observed (Figure 2a). Large oligomers and fibrilar Aβ_1-42_ did not cause necrosis during the 24 h treatment of microglial cultures (Figure 2a). The preparation of Aβ species involves the solvent hexafluoroisopropanol (HFIP), which is evaporated in further steps; however, trace amounts of HFIP in Aβ solutions might be present. Therefore, we performed additional control experiments in which cultures were treated with solvent solutions (HFIP) prepared in exactly the same way as Aβ oligomers but in the absence of peptides. HFIP applied at the same concentration as in the Aβ_1-42_ preparations did not influence the microglial viability (Figure 2a,b).

The data described above show that small Aβ_1-42_ oligomers elevated intracellular Ca^2+^ and induced cell death in pure microglial cultures. Next, we tested the effects of different Aβ_1-42_ preparations on microglia in mixed neuronal–glial cell cultures. After 15 min of incubation, the average microglial calcium-dependent fluorescence in the small Aβ_1-42_ oligomer-treated cultures was 7% higher compared to the untreated control (Figure 3a). After 45 min, the difference from the control increased to 20% and remained similar after 90 min. In contrast, there was no significant difference between the control and monomer-, large oligomer-, and fibril-treated cultures.

The effect of small Aβ_1-42_ oligomers on microglial viability was also observed in mixed neuronal–glial cell cultures. The percentage of necrotic microglia after a 30 min of treatment with small Aβ_1-42_ oligomers was 17% and gradually increased in time to reach 37% after 4.5 h and 53% after 24 h (Figure 3b,d,e). There were almost no necrotic cells in the controls (Figure 3c) and other Aβ_1-42_ species-treated samples. Thus, the levels of cells killed by small oligomers were significantly higher after all investigated periods.

In summary, small Aβ_1-42_ oligomers increased Ca^2+^ levels and induced necrosis in the microglia in pure and mixed neuronal–glial cell cultures; however, the effects were more prominent in pure cultures. No Ca^2+^ or viability changes were observed after treatment with other Aβ_1-42_ forms, such as monomers, large oligomers, or fibrils.

We have previously found that small Aβ_1-42_ oligomers induced neuronal toxicity via NMDARs [8]. In this study, we examined whether NMDARs were involved in microglial Ca^2+^ elevation and cell death triggered by small Aβ_1-42_ oligomers by incubating primary microglial cell cultures with the peptide in the presence of the NMDAR inhibitors MK801, memantine, or D-2-Amino-5-phosphopentanoic acid (DAP5).

All the applied inhibitors at 10 μM concentrations completely prevented the Aβ_1-42_-induced increase in microglial intracellular Ca^2+^ (Figure 4a). In the cells affected by small Aβ_1-42_ oligomers without the inhibitors, the Ca^2+^ level was comparable to the samples treated with a specific NMDAR agonist 1 mM NMDA and was lower than in the cells treated with glutamate, which is an activator of NMDA, AMPA, and kainate receptors. However, the NMDAR inhibitors were not as effective in protecting the microglia from death caused by small Aβ_1-42_ oligomers (Figure 4b). Although the noncompetitive NMDAR inhibitors MK801 and memantine significantly prevented small Aβ_1-42_ oligomer-induced increases in the necrotic cell number after 90, 180, and 270 min of treatment, the levels of dead cells remained substantially higher than in the control cultures (compare Figure 2a,b). A competitive NMDAR inhibitor DAP5 was even less protective and significantly prevented cell death only after 180 and 270 min, but not after 90 min. Neither of the inhibitors significantly affected the microglial viability in the presence of the small Aβ_1-42_ oligomers after 30 min or 24 h. The data suggest that small Aβ_1-42_ oligomers (but not monomers, large oligomers, or fibrils) elevate the intracellular Ca^2+^ concentration in microglia by triggering the NMDARs; however, other mechanisms contribute to this outcome in causing microglial death.

### 2.2. The Effect of Aβ_1-42_ Species on Superoxide Production by Mitochondria

In this part of the study, we investigated whether various Aβ_1_-_42_ species could stimulate mitochondrial superoxide generation in microglia and neurons in mixed neuronal–glial cell cultures. As can be seen in Figure 5a,b, after 30 min of treatment with small Aβ_1-42_ oligomers, superoxide-dependent MitoSOX Red fluorescence in the neuronal cells was 35% higher than in the control, and it remained at this level for 1 h and increased further by 153% after 2 h (Figure 5a,b). Large Aβ_1-42_ oligomers and Aβ_1-42_ fibrils did not cause any detectable changes in the mitochondrial superoxide production. Similarly to the neuronal data, small Aβ_1-42_ oligomers induced a substantial increase in MitoSOX fluorescence in the microglial cells, which reached 195%, 155%, and 167% of the control after 30 min, 1 h, and 2 h of incubation, respectively (Figure 5a,c). Again, other Aβ_1-42_ forms did not significantly influence the mitochondrial superoxide production. In addition, small and large Aβ_1-42_ oligomers or fibrils did not cause any superoxide-related fluorescence in the astrocytes after 0.5–2 h of incubation in neuronal–glial cell cultures. HBSS or 2 μM Antimycin A without MitoSOX did not affect the MitoSOX-related fluorescence in the neurons and microglia.

To get more mechanistic insights into Aβ_1-42_-induced superoxide production by mitochondria, we applied a set of pharmacological modulators. The possible involvement of NMDARs and elevated Ca^2+^ was tested with MK801 and BAPTA, a selective cell-permeable Ca^2+^ chelator. Frentizole was used to identify whether the mitochondrial superoxide was induced by Aβ-ABADs. N-Acetyl L-Cysteine (NAC) has mitochondria-specific antioxidant effects [20] and was applied to examine the importance of the mitochondrial redox state, and Apocynin, an inhibitor of NADPH oxidase, was appliedto verify the role of ROS produced by this enzyme. None of the compounds (except the selective mitochondrial superoxide scavenger MitoTEMPO) affected the small Aβ_1-42_ oligomer-induced neuronal mitochondrial superoxide (Figure 5a,d). However, the microglial Aβ_1-42_-induced mitochondrial superoxide was significantly reduced by the Frentizole, NAC, and BAPTA (and MitoTEMPO), but not by the Apocynin and MK801 (Figure 5e). The results indicate that Aβ-mediated mtROS overproduction is differentially regulated in neurons and microglia in mixed neuronal–glial cultures and point to microglial intracellular Ca^2+^- and ABAD-mediated mitochondrial oxidative stress.

For evaluating whether small Aβ_1-42_ oligomer-induced mtROS cause mitochondrial depolarization and whether it is related to ABAD activation, we measured the mitochondrial membrane potential using MitoTracker Orange CM-H2TMRos in the presence of MitoTEMPO or Frentizole. The exposure of the neuronal–glial cultures to small Aβ_1-42_ oligomers for 1 h caused a 25% decrease in the membrane potential compared to the control (Figure 6a,b). The membrane potential was restored by MitoTEMPO (to 90%) but not with Frentizole (Figure 6a,b), thus suggesting no considerable role of the ABADs in the Aβ-mediated decrease in mitochondrial activity. In addition, MitoTEMPO was not effective in preventing Antimycin A-induced mitochondrial depolarization (Figure 6b), most likely because the latter was caused by respiratory chain complex III inhibition but not by the mtROS. Thus, the result was small Aβ_1-42_-induced mtROS-dependent mitochondrial depolarization in the neuronal–glial cell culture.

### 2.3. The Role of Mitochondrial ROS in Small Aβ_1-42_ Oligomer-Induced Neuronal and Microglial Death

We have previously shown that small Aβ_1-42_ oligomers induce the opening of mitochondrial permeability transition pore in neurons, thereby leading to mitochondrial dysfunction and cell death [8]. Mitochondrial permeability transition pore sensitivity to Ca^2+^ is modulated by mtROS [21], thus suggesting that Aβ_1-42_-affected neurons could be, in principle, rescued by mitochondrially targeted antioxidants or by suppressing ABADs. To test this hypothesis, neuronal–glial cultures were preincubated with MitoTEMPO or Frentizole and then treated with 1 µM small Aβ_1-42_ oligomers for 24 h. Aβ_1-42_ reduced the neuronal viability to 43%, and both Frentizole and MitoTEMPO significantly prevented this loss of viability, thus resulting in 72% and 81%, respectively, of the cells remaining alive (Figure 7a). Twenty-four-hour treatment with small oligomers reduced the microglial viability to 47% of the control (Figure 3b), and the decrease was not MitoTEMPO- or Frentizole-sensitive.

MtROS might modulate glutamatergic signaling leading to calcium overload and excitotoxicity [15]; therefore, we further investigated the potential effects of MitoTEMPO and Frentizole on extracellular glutamate levels in Aβ_1-42_ oligomer-treated neuronal–glial cell culture medium. Indeed, 1 h treatment with small Aβ_1-42_ oligomer elevated the extracellular glutamate from 2.1 µM in the control to 3.9 µM in the treated cultures (Figure 7b). The pretreatment with MitoTEMPO or Frentizole significantly prevented glutamate increases by causing the levels to drop to 2.6 and 2.2 µM, respectively, thereby indicating glutamate release mediation by mtROS and ABAD.

## 3. Discussion

In this study, we used Aβ_1-42_ species of different oligomerization states—small and large oligomers, fibrils, and monomers—to investigate Aβ neurotoxicity mechanisms in rat brain primary cell cultures. The main finding of the study was that the microglial cell culture was extremely sensitive to toxic small Aβ oligomers. Small Aβ_1-42_ oligomers triggered an NMDAR-mediated increase in the intracellular Ca^2+^ in microglial cells, which partially correlated with microglial necrosis. We have previously shown that small oligomers, but not other Aβ forms, induced neuronal necrosis in neuronal–glial cell cultures [22], and now we demonstrate that small oligomers cause the concentration- and time-dependent necrosis of microglial cells. Dystrophic microglia have been found in aged, AD- and frontotemporal-lobar-degeneration-affected brains [23,24,25]. Amyloid plaque-associated microglia have impaired Ca^2+^ signaling and increased apoptosis rates in the AD mouse models [26,27]; however, there are no data about the link between microglial death and neuronal damage in Aβ pathology.

We determined that small Aβ_1-42_ oligomers caused mtROS, mitochondrial depolarization, high extracellular glutamate levels, and neuronal death in neuronal–glial cell cultures; and mitochondrial antioxidants prevented all these events. Aβ-induced mtROS overproduction in microglia was mediated by intracellular Ca^2+^ and ABADs. The interaction of Aβ with ABAD enhances mitochondrial oxidative stress, mitochondrial toxicity, and cognitive decline in AD patients and transgenic mice [16], whereas the inhibition of ABAD–Aβ interaction reduces oxidative stress, as well as improves mitochondrial functions and spatial memory in a mouse AD model [28]. Several studies indicate that the molecular pathways under redox control may be implicated in the Aβ pathogenesis targeting microglia [29]. In our study, Apocynin was not effective in the Aβ-mediated ROS production in microglia, and this result suggests that mtROS, not NADPH oxidase derived-ROS, might be responsible for mediating microglial damage in response to small Aβ oligomers. 

Mitochondrial dysfunction in microglia plays a significant role in the pathogenesis of AD and other neurological disorders [30]. Previously, we found that small Aβ_1-42_ oligomers decreased the capacity of isolated brain mitochondria to retain calcium, thus inducing mitochondrial permeability transition pore opening [8]. The overload of calcium in mitochondria leads to the opening of mitochondrial permeability transition pore, which is an initial step in activating necrotic and apoptotic cell death [31,32]. Moreover, ROS overproduction in neurons and glia can serve as an additional trigger for pore opening [33]. One of the possible causes of enhanced mtROS might be the direct impairment of complex I activity by Aβ [34].

Here, the Aβ_1–42_ oligomers caused a rapid increase in the NMDAR-dependent cytosolic calcium in microglia and a calcium-dependent increase in the mtROS and extracellular glutamate, thereby inducing neuronal and microglial death. The overproduction of mtROS in neurons alters the NMDARs, thus leading to impaired glutamatergic signaling, calcium overload, and excitotoxicity [15]. Oxidative glutamate toxicity has been hypothesised as the component of excitotoxicity-initiated cell death when parts of the neurons are killed by glutamate exposure directly in an antioxidant-dependent manner, and another part is damaged due to the activation of the NMDARs [35]. Aβ_1-42_ oligomers stimulated the excessive formation of mtROS through a mechanism requiring NMDAR activation in hippocampal neuronal cultures [36]. Ionotropic glutamate receptors are also expressed in microglial cells, and their stimulation leads to a Ca^2+^ influx [37] and ROS production [38]. In addition, glutamate might cause an extracellular Ca^2+^ influx via Na^+^/Ca^2+^ exchanger [39].

One of the possible explanations for NMDAR-mediated mitochondrial response induced by Aβ oligomers could be the involvement of mitochondria-associated membrane-dependent NMDARs. It has been shown that the activation of NMDARs may trigger efficient Ca^2+^ release from endoplasmic reticulum (ER) stores [40,41], and the NMDA receptor antagonist DAP5 may suppress ER Ca^2+^ release and mitochondrial Ca^2+^ import in mammalian neurons [42]. Earlier investigations on postmortem brains affected by AD have determined the increase of contact sites between the ER and mitochondria, as well as the upregulation of proteins of the mitochondria-associated membranes [43]. Additionally, Aβ-induced NMDAR activation can cause mitochondrial dysfunction via Ca^2+^ flux from the ER in primary cortical neurons [44]. Thus, the activation of the NMDAR is required for Ca^2+^ release from ER stores and the Ca^2+^ import into neuronal mitochondria. However, it is not known whether NMDARs are important for ER/mitochondria Ca^2+^ fluxes in microglial cells.

Our findings suggest that the downstream effects of small Aβ oligomers leading to microglial calcium overload and NMDAR activation, microglial and neuronal mitochondrial damage, and glutamate release may contribute to the mitochondrial oxidative stress hypothesis and may provide a basis for the development of therapeutic measures for delaying the progression of AD. The effects of small Aβ oligomer-induced toxicity on neurons and microglia, as well as their potential pharmacological modulation sites, are summarized in a scheme in Figure 8.

## 4. Materials and Methods

### 4.1. Materials

Synthetic Aβ_1-42_ was obtained from Bachem (Bubendorf, Switzerland), Isolectin GS-IB4 from Griffonia simplicifolia conjugated with Alexa Fluor488, MitoSOX™ Red, MitoTracker Green FM dye, MitoTracker^®^ Orange CM-H2TMRos, Amplex red glutamic acid/glutamate oxidase assay kit, and TNF-α (Rat) ELISA kit, were purchased from Invitrogen, ThermoFisher Scientific (Waltham, MA, USA); glucose (ROTH), penicillin–streptomycin (Biochrom, Cambridge, UK), Versene was obtained from Gibco (Paisley, UK). All other materials were purchased from Sigma-Aldrich, Burlington, MA, USA.

### 4.2. Preparation of Aβ_1-42_ Aggregates

Aβ_1-42_ species, monomers, small oligomers, large oligomers, and fibrils were prepared as described in Cizas et al. (2010) [22]. The average size of small Aβ_1-42_ oligomers was below 5 nm z-high and >5 nm for large Aβ_1-42_ oligomers, as measured by atomic fluorescence microscopy (AFM) [22]. HFIP was used as a solvent to prepare all Aβ_1-42_ species. Vehicle (referred as HFIP) was prepared in the same way as oligomeric forms but without Aβ.

### 4.3. Cell Cultures and Treatments

The procedures used in this study were approved by The State Food and Veterinary Service of the Republic of Lithuania in accordance with European Convention for the protection of vertebrate animals used for experimental and other purposes. The rats were bred and maintained at Lithuanian University of Health Sciences Animal House under controlled conditions. Wistar rats were anaesthetised by CO_2_, followed by cervical dislocation. Primary pure microglial cultures from rat cerebral cortices were prepared from 7–8-day-old Wistar rats of both genders as described in [45].

Primary neuronal–glial cultures from rat cerebellum were prepared from 7–8-day-old Wistar rats as described in [46]. Cells were grown for 7 days in vitro (DIV) before exposure to Aβ species. The neuronal–glial culture contained 80% neurons and 7% astrocytes as assessed by cellular morphology and 13% microglia indicated by staining with Isolectin GS-IB4 conjugated with Alexa Fluor488.

### 4.4. Measurement of Intracellular Calcium Concentration

Intracellular calcium concentration was assessed by fluorescence microscopy using the Fluo-3AM dye. Cultures were exposed to Aβ_1-42_ compounds (monomers, small oligomers, large oligomers, and fibrils), at 1.0 μM, and Fluo-3AM at 2.0 μM for 15, 45, and 90 min; all were visualized by OLYMPUS IX71S1F-3 fluorescent microscope (at 495/521 nm wavelengths). To investigate the role of NMDARs, the cell cultures were preincubated with NMDAR blockers (memantine, MK801, and DAP5) all at 10 µM for 30 min. A total of 1 mM ATP-induced Fluo-3AM fluorescence was evaluated as maximal and was considered 100%. Fluo-3AM fluorescence after treatment with BAPTA (6 µM) was considered minimal (0%). For calcium evaluation in microglia in neuronal–glial cultures, the cells were treated and assessed the same way as in pure culture, and fluorescence intensity was measured in individual microglial cells that were traced according to their brightfield images, as demonstrated in Figure 9.

### 4.5. Cell Viability Assay

The viability of cells in cultures was assessed by propidium iodide (PI, 7 µM) and Hoechst 33342 (4 µg/mL) staining using a fluorescence microscope OLYMPUS IX71S1F-3, as described in [22]. Neurons were recognised according to characteristic morphology (round in shape, small somata with a few dendrites) using phase-contrast microscopy. PI-positive cells were classified as necrotic, and cells with condensed/fragmented nuclei (Hoechst 33342, bright blue) as apoptotic. Microglial cells were identified by Isolectin GS-IB4 conjugated with AlexaFluor488 (7 ng/mL) staining. Neuronal and microglial cell numbers in neuronal–glial cultures were assessed by counting cells in at least 5 microscopic images/wells by means of ImageJ software. Cell viability was expressed as the percentage of viable, necrotic, or apoptotic cells of the total number of specific cells in the microscopic image. The number of neurons/microglia in the control group was considered as 100%. Cultures were treated with Aβ_1-42_ species (0.5 or 1 µM) in the presence/absence of NMDAR blockers memantine, MK801, or DAP5. The blockers were applied at 10 µM concentrations and added 30 min before the Aβ treatment. MitoTEMPO (10 μM) or Frentizole (5 μM), where applied, were added 15 min before the Aβ treatment. List of inhibitors used in the study is indicated in Table 1.

### 4.6. Mitochondrial Superoxide Measurements

Cultures were incubated with MitoSOX Red (200 nM) and MitoTrackerGreen (200 nM) for 30 min at 37 °C, washed by Hanks’ Balanced Salt Solution (HBSS) and analyzed by fluorescence microscopy OLYMPUS IX71S1F-3 (495/521 nm and 536/617 nm bandpass filters). For evaluating the effects of Aβ_1-42_, neuronal–glial cultures were treated with various Aβ_1-42_ species (1 µM) for 0.5, 1, or 2 h. In some cases, culture was pretreated with MK801 (10 µM), APO (1 mM), NAC (100 µM), BAPTA (5 µM), MitoTEMPO (10 µM), or Frentizole drug (5 μM) for 15 min before addition of Aβ_1-42_. For positive control, the cells were treated with Antimycin A, in concentration of 2 µM, for 30 min. Images of at least 5 randomly selected fields per well were taken, and the changes in red fluorescence were evaluated and analyzed by Image J 1.52 v software.

### 4.7. Mitochondrial Membrane Potential Assessment

Mitochondrial membrane potential in neuronal–glial cultures was monitored using the fluorescent dye MitoTracker^®^ Orange CM-H2TMRos (200 nM) according to the manufacturer protocol. Cultures were treated with small Aβ_1-42_ oligomers plus/minus MitoTEMPO or Frentizole, as described in Section 4.5. The ability of the dye to respond to plasma membrane potential changes was tested by a mitochondrial membrane potential destabilizer Antimycine A (2 µM) for 30 min. Pictures of at least 5 randomly selected fields per well were taken using fluorescence microscope OLYMPUS IX71S1F-3 with 554/576 nm bandpass filters. The changes in red fluorescence were analyzed by Image J 1.52 v software and calculated per total number of cells.

### 4.8. Measurement of Glutamate Concentration

Extracellular glutamate was measured using the commercially available Amplex Red glutamic acid/glutamate oxidase assay kit according to the manufacturer’s protocol. Neuronal–glial cultures were pretreated with small Aβ_1-42_ oligomers for 1 h plus/minus MitoTEMPO or Frentizole. After incubation, 50 µL aliquots of culture medium were taken and mixed with a working solution of 100 µM Amplex Red reagent containing 0.25 U/mL HRP, 0.08 U/mL L-glutamate oxidase, 0.5 U/mL L-glutamate–pyruvate transaminase, and 200 µM L-alanine; they were then incubated for 30 min at 37 °C and protected from light. The fluorescence was measured in a fluorescence microplate reader Fluoroskan Ascent (Thermo Scientific, Waltham, MA, USA) using excitation in the 530–560 nm range and emission at 590 nm. Glutamate concentrations of the test samples were calculated from the L-glutamic acid standard curve.

### 4.9. Statistical Analysis

Data were expressed as mean ± standard deviation SD of 4–9 experiments on separate cultures. Statistical comparison between experimental groups was performed using a one-way ANOVA followed by a Tukey’s or LSD post hoc test using SPSS 20.0 software. *p* values < 0.05 were considered significant.

## Figures and Tables

**Figure 1 ijms-24-12315-f001:**
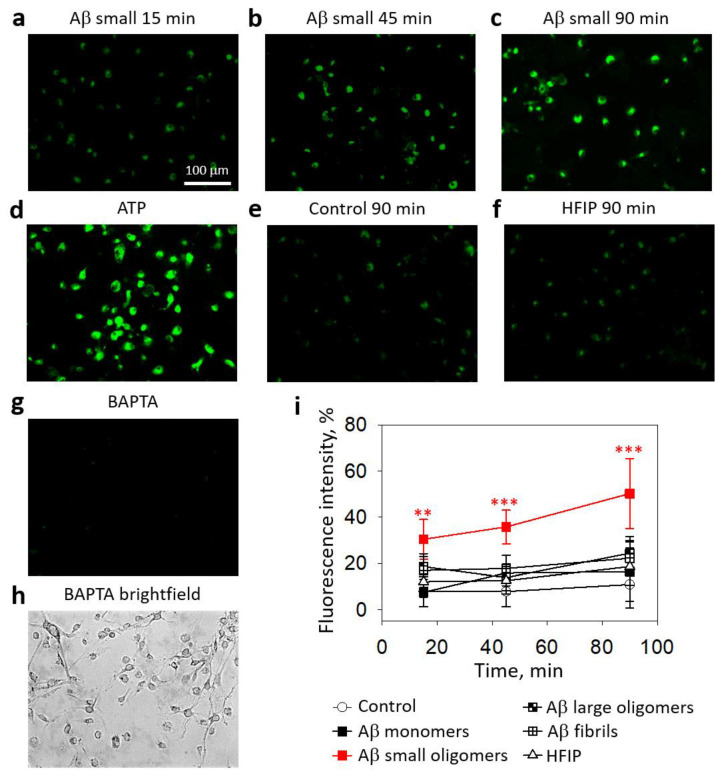
The effect of different Aβ_1-42_ species on Ca^2+^ concentration in microglial cells:Intracellular Ca^2+^ was assessed by loading microglia with Fluo-3AM dye and measuring the fluorescence intensity of microscope images. In (**a**–**h**), there are representative images of the cells; the scale bar (100 µm) is the same for all images. In (**i**)—a quantitative evaluation of Ca^2+^-dependent fluorescence in microglial cells. HFIP—hexafluoroisopropanol is a solvent used for Aβ preparation. BAPTA—1,2-Bis(2-aminophenoxy)ethane-N,N,N′,N′-tetraacetic acid tetrakis(acetoxymethyl ester) was used as a negative control of the assay. ATP was used as a positive control. The fluorescence intensity in each experiment was expressed as a percentage of ATP-stimulated fluorescence, and BAPTA-treated cell fluorescence level was eliminated as a nonspecific baseline. The data are presented as averages with a standard deviation of 5–9 experiments performed in triplicates (5 images each, i.e., 15 images per sample in total). ***, **—statistically significant difference compared with the HFIP control at the same time point. ***—*p* < 0.001, **—*p* < 0.01.

**Figure 2 ijms-24-12315-f002:**
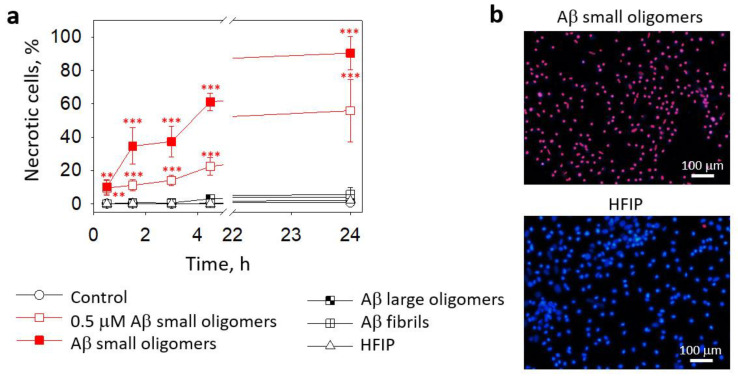
The effect of Aβ_1-42_ peptide on microglial viability in pure microglia cultures: In (**a**), there are average percentages of necrotic (PI-positive) nuclei in fluorescence microscope images taken after treatment with Aβ_1-42_ peptides of different aggregation states (0.5 µM and 1 µM small Aβ small oligomers; 1 µM Aβ large oligomers and fibrils). HFIP—a solvent used for Aβ preparation. In (**b**), there are representative images of microglial nuclei stained for viability detection with Hoechst33342 and PI after 24 h treatment with the small Aβ_1-42_ oligomers. Cell cultures were analyzed at 20X magnification. ***, **—statistically significant difference compared with the control at the same time point; ***—*p* < 0.001; **—*p* < 0.01.

**Figure 3 ijms-24-12315-f003:**
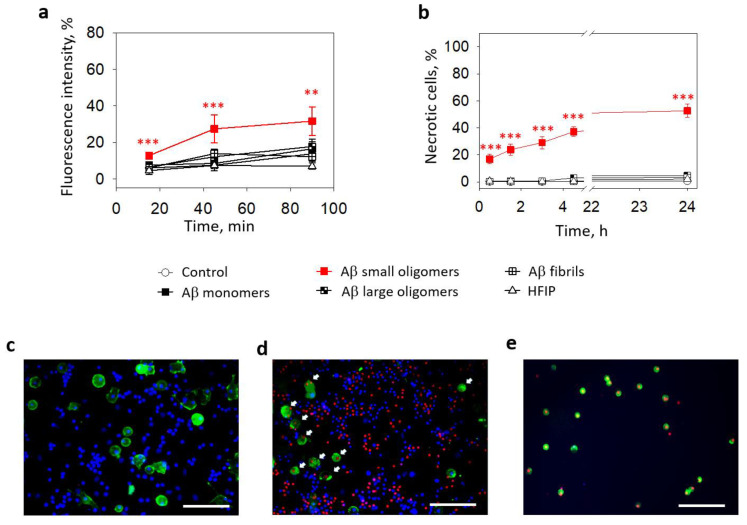
The effect of Aβ_1-42_ peptide on microglial viability in neuronal-glial cultures: In (**a**), microglial Ca^2+^-dependent Fluo 3AM fluorescence intensity after treatment with with Aβ_1-42_ peptide of different aggregation states. In (**b**), there are average percentages of necrotic (PI-positive) nuclei in fluorescence microscope images taken after treatment with Aβ_1-42_ peptides of different aggregation states. HFIP—a solvent used for Aβ preparation. In (**c**), there is a representative image of solvent (HFIP-treated) control samples of neuronal–glial cell cultures. In (**d**)—24 h small Aβ_1-42_ peptide oligomer-treated sample of neuronal–glial cell culture. In (**e**)—surface-floating microglial cells after 24 h small Aβ_1-42_ peptide oligomer treatment. Microglial cells are stained green with Isolectin B4; all nuclei are stained for viability detection with Hoechst 33342 and PI. White arrows in (**d**) point to the microglial cells with PI-positive necrotic nuclei. All the quantitative data are presented as averages with a standard deviation of 5–7 experiments performed in technical triplicates; 5 images each (15 images per sample per experiment in total). The scale bar in (**c**–**e**) is 100 μm. ***, **—statistically significant difference compared with the control at the same time point; ***—*p* < 0.001; **—*p* < 0.01.

**Figure 4 ijms-24-12315-f004:**
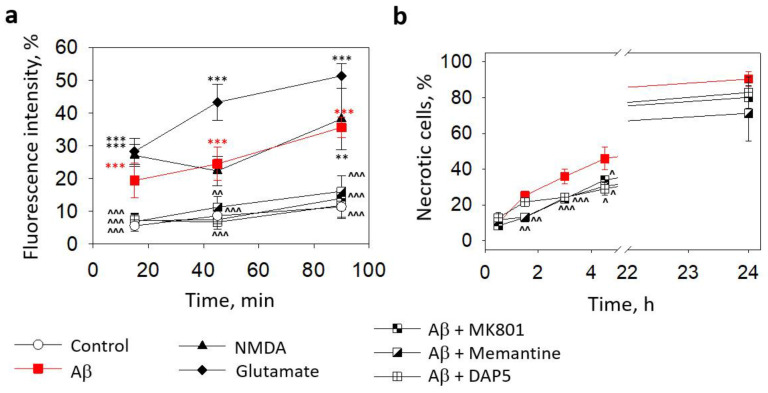
The effect of Aβ_1-42_ peptide on microglial viability in pure microglia cultures and involvement of NMDARs in Aβ-induced intracellular Ca^2+^ elevation and toxicity: In (**a**), microglial Ca^2+^-dependent Fluo 3AM fluorescence intensity after treatment with small Aβ_1-42_ oligomers in the absence and presence of NMDAR inhibitors (MK801, memantine, and DAP5, all at 10 µM concentration). In (**b**), the effect of NMDAR inhibitors on the level of microglial death induced by small Aβ_1-42_ oligomers. DAP5—D-2-Amino-5-phosphopentanoic acid. All the quantitative data are presented as averages with a standard deviation of 5–7 experiments performed in technical triplicates, 5 images each (15 images per sample per experiment in total). ***, **—statistically significant difference compared with the control at the same time point; ^^^, ^^, ^—compared with Aβ. ***, ^^^—*p* < 0.001; **, ^^—*p* < 0.01; ^—*p* < 0.05.

**Figure 5 ijms-24-12315-f005:**
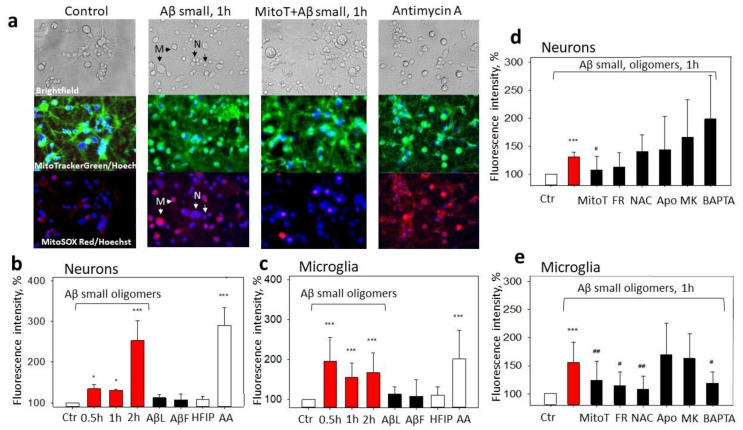
The effect of Aβ_1-42_ peptide on mitochondrial superoxide production in neuronal–glial cultures: In (**a**), there are representative images of the control cells; cells pretreated with small Aβ_1-42_ oligomers; cells pretreated with MitoTEMPO + small Aβ_1-42_ oligomers; and cells pretreated with Antimycin A, upper line—phase contrast light microscopy images; middle line—cells stained using green fluorescent MitoTrackerGreen dye and Hoechst 33342; bottom line—images of cells stained with MitoSOX Red and Hoechst 33342. In (**b**), there are average percentages of MitoSOX Red fluorescence intensity after treatment with various Aβ_1-42_ species (small Aβ oligomers, large oligomers, (AβL), Aβ fibrils (AβF)) at 1 µM concentration in neuronal cells; (**c**) in microglial cells. HFIP—a solvent used for Aβ preparation. AA—Antimycin A (2 µM), a positive control. In (**d**), there are average percentages of MitoSOX Red fluorescence intensity after treatment with small Aβ_1-42_ oligomers plus various inhibitors in neuronal cells; (**e**) in microglial cells. Cultures were pretreated with MitoTEMPO (MitoT) (10 µM), Frentizole drug (FR) (5 μM), N-Acetyl Cysteine (NAC) (100 µM), Apocynin (Apo) (1 mM), MK801 (MK) (10 µM), and BAPTA (5 µM), see Section 4. All data are presented as averages with a standard deviation of 4-8 experiments performed in technical triplicates, five images each (15 images per sample per experiment in total). The changes in red fluorescence were evaluated in neuronal and microglial cells separately. The fluorescence intensity was measured in individual cells that were traced according to their bright field/fluorescence images; marked with arrows—neurons, N, and microglia, M. Cell cultures were analyzed at 20× magnification. ***, *—statistically significant difference compared with the control (Ctr); #, ##—with Aβ; ***—*p* < 0.001; ##—*p*<0.01, *, #—*p* < 0.05.

**Figure 6 ijms-24-12315-f006:**
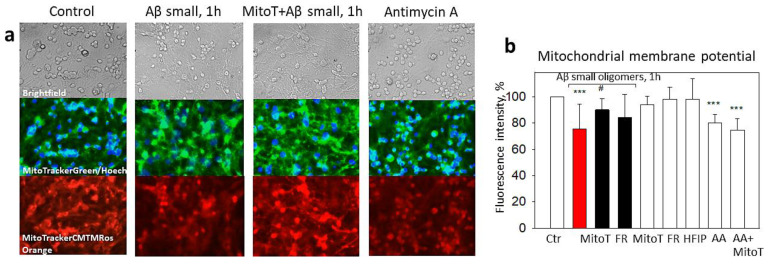
The effect of small Aβ_1-42_ oligomer-induced mitochondrial membrane potential collapse in neuronal-glial cultures: Mitochondrial membrane potential was monitored using fluorescent dye MitoTracker Orange CM-H2TMRos. In (**a**), there are representative images of the control cells; cells pretreated with small Aβ_1-42_ oligomers; cells pretreated with MitoTEMPO + small Aβ_1-42_ oligomers and cells pre-treated with Antimycin A; upper line—phase contrast light microscopy images; middle line—cells stained using green fluorescent MitoTrackerGreen dye and Hoechst 33342; bottom line—images of cells stained with MitoTracker Orange CM-H2TMRos. (**b**)—a quantitative evaluation of mitochondrial membrane potential-dependent fluorescence in cultures pretreated with MitoTEMPO (MitoT), Frentizole (FR), HFIP or Antimycin A (AA). All the quantitative data are presented as averages with a standard deviation of 4–8 experiments. Cell cultures were analyzed at 20× magnification. ***—statistically significant difference compared with the control; #—with Aβ; ***—*p* < 0.001; #—*p* < 0.05.

**Figure 7 ijms-24-12315-f007:**
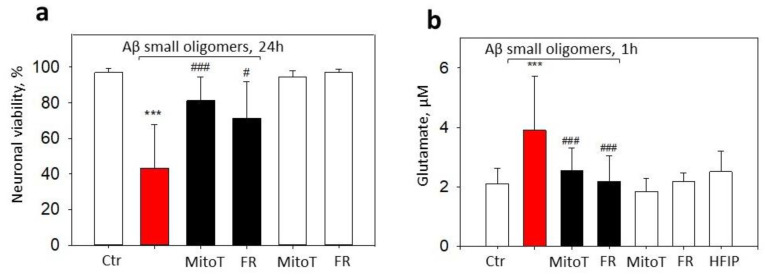
The effect of MitoTEMPO and Frentizole on small Aβ_1-42_ oligomer-induced neuronal death and extracellular glutamate in neuronal-glial cultures: (**a**)—viable neurons; (**b**)—glutamate. In cell death experiments (**a**), cells were exposed to 1 µM Aβ_1-42_ oligomers with or without MitoTEMPO (MitoT) or Frentizole (FR) for 24 h. Cell death was quantified by assessing nuclei morphology after staining with PI and Hoechst 33342. Cells were counted in at least 5 microscopic fields per well (two wells per treatment). Data are expressed as percentage of specific neuronal cells of the total number of neuronal cells per field. Neurons were recognized according to characteristic morphology using phase contrast microscopy. For extracellular glutamate measurements (**b**), cells were treated with 1 µM small Aβ_1-42_ oligomers for 1 h with or without MitoTEMPO or Frentizole. Glutamate in culture medium was evaluated by Amplex Red glutamic acid/glutamate oxidase assay kit. Glutamate concentrations were calculated with L-glutamic acid standard curve. All the quantitative data are presented as averages with a standard deviation of 4–8 experiments. ***, statistically significant difference compared with the control; ###—with Aβ; #—*p* < 0.05; ***, ###—*p* < 0.001.

**Figure 8 ijms-24-12315-f008:**
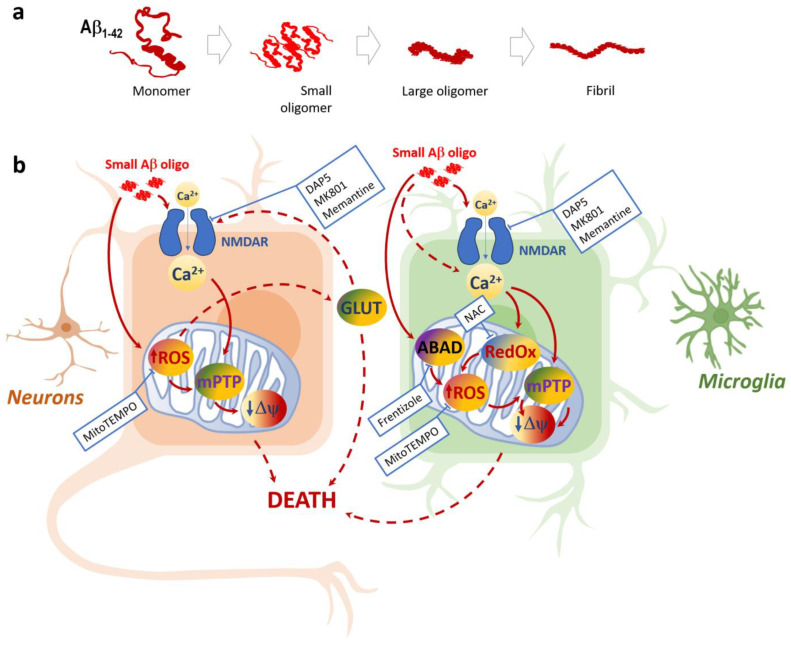
Aggregation states of Aβ_1-42_ peptide (**a**) and schematic representation of small Aβ_1- 42_ oligomer-induced toxicity in neuronal and microglial cells (**b**): Only small Aβ_1-42_ peptide oligomers are toxic to neurons and microglia; the toxicity mechanisms in these two cell types share some similarities but also have differences. Both microglia and neurons suffer from NMDA-dependent Ca^2+^ increases followed by mitochondrial ROS increases, as well as the loss of mitochondrial inner membrane potential ΔΨ (most likely due to the Ca^2+^ and ROS-induced mitochondrial permeability transition pore mPTP). The mitoROS feedback to NMDARs via glutamate (GLUT) increase can lead to neuronal death due to excitotoxicity and deenergization. In microglia, NMDAR is not the only pathway of Aβ_1-42_-induced Ca^2+^ entry, and the importance of ABAD and the mitochondrial redox state is evident. Small Aβ_1-42_ oligomer-mediated damage in neurons and microglia can be pharmacologically controlled by NMDAR inhibitors MK801, memantine, and DAP5, as well asmitochondrial ROS scavenger MitoTEMPO. In microglia, this damage can be prevented by ABAD inhibitor Frentizole and mitochondrial redox state modulator N-Acetyl L-Cysteine (NAC).

**Figure 9 ijms-24-12315-f009:**
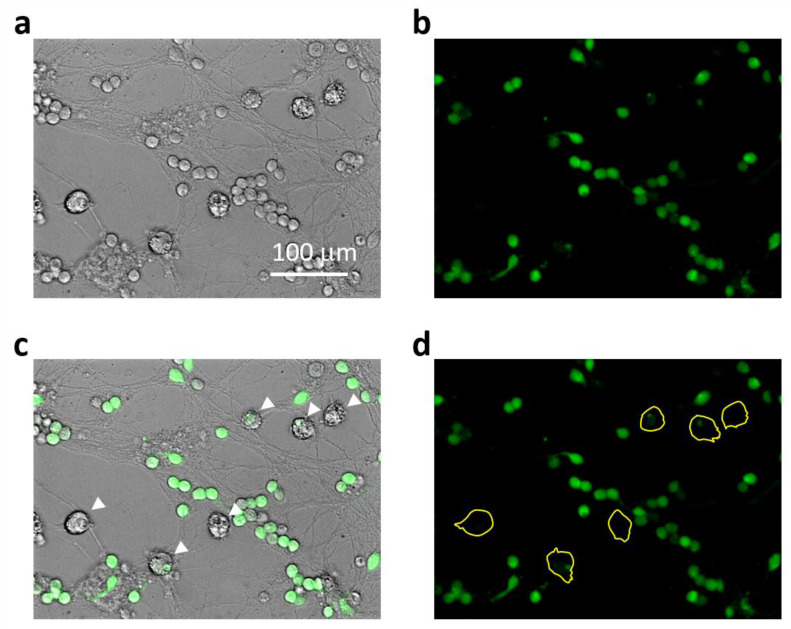
Representative images explaining Ca^2+^-related fluorescence intensity assessment in microglial cells in mixed neuronal-glial cultures: In (**a**), there is a brightfield phase-contrast image of the culture after 45 min treatment with small Aβ_1-42_ oligomers. Small round bodies of about 10–12 μm in diameter belong to cerebellar granule neurons and those larger ones of about 20 μm belongto microglial cells. Astrocyte cells form a layer below the neurons and are not so well visible under phase contrast. In (**b**), the same cells visualized for Ca^2+^-dependent fluorescence of Fluo-3AM preloaded to the cells before the Aβ_1-42_ treatment. In (**c**), the brightfield image merged with the fluorescent one to indicate the Ca^2+^-related fluorescence in microglial bodies (indicated by white arrows). In (**d**), the fluorescent image areas covered by microglial bodies were traced yellow to evaluate the fluorescence intensity.

**Table 1 ijms-24-12315-t001:** Inhibitors used in the study.

Inhibitor	Mechanism of Action
MK801	A potent noncompetitive NMDAR antagonist [47].
Memantine	A low-affinity noncompetitive NMDAR antagonist [48].
D-2-Amino-5-phosphopentanoic acid	A competitive NMDAR antagonist [47].
BAPTA	A selective cell-permeable Ca^2+^ chelator (information provided by manufacturer).
MitoTEMPO	A mitochondrially targeted antioxidant and specific scavenger of mitochondrial superoxide (information provided by manufacturer).
Frentizole	An inhibitor of amyloid beta peptide binding alcohol dehydrogenase (ABAD)–Aβ interaction (information provided by manufacturer).
N-Acetyl L-Cysteine	A disulfide reductant, a direct scavenger of oxidants, and a driver of glutathione synthesis [20,49].
Apocynin	A potent and selective inhibitor of NADPH oxidase [50].
Antimycin A	An inhibitor of mitochondrial cytochrome bc1 (complex III), often used as a positive control for superoxide generation.

## Data Availability

The raw data supporting the conclusions of this manuscript will be made available by the authors, without undue reservation, to any qualified researcher.

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
