# Peer review of "The Role of Intracellular Ca^2+^ and Mitochondrial ROS in Small Aβ_1-42_ Oligomer-Induced Microglial Death"

_ijms, 2023, doi:10.3390/ijms241512315_

Round 1
Reviewer 1 Report
Major points:
Figure 9: a is the same picture as c. So it can not be considered a control.
b is the same picture as d. So it can not be considered d control.
You should consider changing the pictures.
Can you think that there are other receptors involved in the death of the microglia? Can you tell me more about them? For example the Calcium Sensing Receptor.
Why small oligomers are so effective to kill rat microglia in humans this effect is not so evident. A rat treated with these small oligomers will die in a few days.
In your previous work, it appeared that microglia did not die with the small oligomers. Can you explain it? See your reference 9.
In Rat microglia, the activation of NADPH oxidase requires that beta-amyloid peptides be in the fibrillary state, is inhibited by inhibitors of tyrosine kinases or phosphatidylinositol 3-kinase and by dibutyryl cyclic AMP, and is potentiated by interferon-gamma or tumor necrosis factor-alpha.
Check your references there are some errors, and numbers not matched with the references, see lane 144. (8-9)
Figure 5: the legend is quite difficult to read you should improve it.
Reviewer 2 Report
The role of different Ab42 species (monomers, oligomers, fibrils etc) on cell toxicity and on mitochondrial function remains ambiguous.
The authors of the present study chose to these different Ab42 species in primary microglial and neuronal cultures and explored potential mechanisms using selective inhibitors.
Overall, the manuscript was well written (although some passages were a bit long-winded and could use some editing) and the experiments were straightforward. The data appear to support their working hypothesis. The authors concluded that much of the mitochondria-mediated toxicity associated with Ab42 appears to relate to microglia-dependent mechanisms (and not so much neurons) and appears to be caused primarily by the smaller oligomeric species (versus larger, more complex conformations).
General comments:
I do not have issue with the way the manuscript presents the research question and the supporting literature. I would have liked a sentence or two on the possibility that the NMDAR-mediated effects may not be through the plasma membrane, but due to effects in closer proximity to the mitochondria (e.g. an MAM-dependent NMDAR population), but this is not a fatal flaw.
Some minor comments:
1) The cell cultures used are not described (except in the Mat/Mets section), so could use a descriptor in the text itself, i.e. line 67 could be re-worded to state: ‘primary microglial cultures’…same with neurons in line 111 (and again in line 224);
2) ‘Treatment’ and ‘Control’ (lines 103 and 107) do not need to be capitalized (I haven’t checke, but this would apply anywhere else in the text);
3) Figure 1, legend: ‘different Ab1-42 species (not peptides);
4) Figure 3: do not need to say ‘intramicroglial’…’microglial’ suffices;
5) Figure 5: the authors start referring to Ab oligomers as Abo…whereas up to that point they used the full term. I would stick with the full term for sake of consistency.
6) Figure 8: replace ‘degrees’ with ‘states’ or ‘conformations’ (and anywhere else in the text where ‘degrees’ is used in this same context.
7) All of the figures are very grainy and need to be replaced with higher resolution images;
8) I am not quite sure how the authors used HFIP…was this used for every Ab42 species solution or just for the small oligomers? HFIP will denature the larger complexes, rendering them useless (or breaking them down to smaller complexes depending on the duration or concentration of HFIP). This needs to be clarified.
9) Given the large number of inhibitors used in the study, the manuscript could benefit from a summary table of the inhibitors and their supposed mechanism of action;
10) I would avoid using the term ‘MPTP’ as it could distract any reader into thinking you are using the parkinsonism-inducing drug (gets converted to MPP+ and affects mitochondria). I would just use the full term, i.e. ‘mitochondrial permeability transition pore’, as it is only used infrequently in the text.
11) Finally, all of the references are out by one position, i.e. ref 1 in the ref list is a blank and ref 2 in the list is actually ref 1 in the text and so on…needs to be fixed.
FIne
Round 2
Reviewer 1 Report
The authors respond to the questions that arise from their work, some questions remain to be answered but the manuscript can now be published.